# Characterization of Hot Deformation of near Alpha Titanium Alloy Prepared by TiH_2_-Based Powder Metallurgy

**DOI:** 10.3390/ma15175932

**Published:** 2022-08-27

**Authors:** Rongxun Piao, Wenjin Zhu, Lan Ma, Peng Zhao, Biao Hu

**Affiliations:** 1School of Mechanical Engineering, Anhui University of Science and Technology, Huainan 232001, China; 2Department of Vanadium and Titanium, Panzhihua University, Panzhihua 617000, China; 3School of Materials Science and Engineering, Shandong Jianzhu University, Jinan 250101, China; 4Anhui International Joint Research Center for Nano Carbon-Based Materials and Environmental Health, Huainan 232001, China

**Keywords:** near-α titanium alloy, TiH_2_-based powder metallurgy, hot compression, strain hardening exponents, strain rate sensitivity, processing map

## Abstract

TiH_2_-basd powder metallurgy (PM) is one of the effective ways to prepared high temperature titanium alloy. To study the thermomechanical behavior of near-α titanium alloy and proper design of hot forming, isothermal compression test of TiH_2_-based PM near-α type Ti-5.05Al-3.69Zr-1.96Sn-0.32Mo-0.29Si (Ti-1100) alloy was performed at temperatures of 1123–1323 K, strain rates of 0.01-1 s^−1^, and maximum deformation degree of 60%. The hot deformation characteristics of alloy were analyzed by strain hardening exponent (*n*), strain rate sensitivity (*m*), and processing map, along with microstructure observation. The flow stress revealed that the difference in softening/hardening behavior at temperature of 1273–1323 K and the strain rate of 1 s^−1^ compared to the lower deformation temperature and strain rate. The strain hardening exponents at temperatures of 1123 K are all negative under all strain rates, and the most severe flow softening with minimum value of *n* was observed at 1123 K and 1 s^−1^. The strain rate sensitives showed that the peak region with *m* value greater than 0.5 generally appeared in the high temperature range of 1273–1323 K, while strain rate sensitivity at low temperature behaved differently with strain rates. The processing map developed for strain of 0.6 exhibited high power dissipation efficiency at high temperatures of 1273–1323 K and a low strain rate of 0.01 s^−1^, due to microstructure evolution of *β* phase. The decrease of strain rate at 1323 K resulted in the formation of globularization of α lamellae. The instability domain of flow behavior was identified in the temperature range of 1123–1173 K and at the strain rate higher than 0.01 s^−1^ reflecting the localized plastic flow and adiabatic shear banding, and inhomogenous microstructure. The variation of power dissipation energy (*η*) slope with strain demonstrated that the power dissipation mechanism during hot deformation has been changed from temperature-dependent to microstructure-dependent with the increase of temperature for the alloy deformed at 0.1 s^−1^. Eventually, the optimum processing range to deform the material is at 1273–1323 K and a strain rate range of 0.01–0.165 s^−1^ (ln$\dot{\varepsilon }$ = −4.6–−1.8).

## 1. Introduction

Near-α titanium alloys are widely used in compressor discs and blades of jet engines because of their excellent high-temperature fatigue and creep properties, high strength-to-weight ratio, and good corrosion resistance [1]. Ti-1100 (Ti-6Al-2.75Sn-4Zr-0.4Mo-0.4Si, wt%) alloy has the highest capability to tolerate temperature up to 873 K without any degradation in mechanical properties among near-α titanium alloys [2]. Like most titanium alloys, near-*α* titanium alloy has poor formability and high cost, which has always been two key problems in a wide range of commercial applications [3]. Powder metallurgy (PM) is one of the most promising methods for production of titanium alloy components at low cost [4]. However, compared with ingot metallurgy (IM) titanium alloys, challenge of PM titanium alloy are high porosity and high oxygen contamination, which usually lead to detrimental effect on the final product by lowering ductility or other defects [5]. 

One of the effective ways to solve the two problems of insufficiently high density and oxygen contamination is to use TiH_2_ powder as the main raw material. It has been demonstrated previously that PM parts developed using TiH_2_ powder feedstock had better control over lower oxygen content, the microstructures, and good chemical homogeneity [6,7,8]. Using TiH_2_ powder, density larger than 95% and good performance with fine grain microstructure can be easily obtained [6,8,9]. Previously, Hagiwara et al. [10] have prepared Ti-1100 alloy with ultra low chlorine hydride-dehydride (ELCL-HDH) titanium powder and master alloy powder as main raw materials through the means of blended elemental (BE) PM synthesis of cold pressing-vacuum sintering-heat treatment-hot isostatic pressing, and the prepared alloys have high density of 95% and better performance in tensile strength and high cycle fatigue strength compared to the conventional PM process. Zhang et al. [3] investigated the mechanical behavior of TiH_2_-based near-*α* Ti–3Al–2Zr–2Mo titanium alloy prepared by a series process of cold pressing-sintering-hot extrusion-vacuum annealing-common annealing, and the prepared PM alloy samples showed a satisfying combination of superior tensile strength and excellent ductility. Using similar methods, recently, Zhang et al. [11] and Wu et al. [12] in the same research group studied the tensile properties of a hot-extruded TiH_2_-based near-α Ti-3Al-2Zr-2Mo-0.36O and Ti-6Al-2Sn-titanium alloy, and the prepared material has shown high performance of tensile ductility. 

For the proper design of thermal deformation of metal, it is necessary to study the thermomechanical behavior. The mechanism of flow stress softening as well as strain hardening phenomena determines the thermomechanical behavior of a material, depending on processing parameters such as strain, strain rate, and temperature. For the description of the hot deformation behavior of titanium alloys, the dynamic material model (DMM) [12,13] based processing map technology has been widely used to provide the optimization of processing parameters and processing windows with instability regions [14]. Likewise, the strain hardening, strain rate sensitivity, and temperature sensitivity of the alloys during deformation can provide the reference and guidance for better thermal deformation of the alloy. Previously, many studies have focused on the hot deformation of as-cast near-α titanium alloys. For example, Krishna et al. [15] investigated the hot deformation mechanism in near α titanium alloy 685 by constructing a processing map, obtaining the optimal condition of 1248 K and 0.001 s^−1^, and proposing that dynamic recrystalization (DRX) occurred due to low oxygen level present in the alloy. Balasundar et al. [16] discussed geometric dynamic recrystallization behavior of a near-α titanium alloy TTTAN 29A (equivalent to IMI834) with acicular microstructure through hot compression tests at temperatures of 1123–1333 K and a strain rate of 3 × 10^−4^–1 s^−1^, and proposed the optimum condition is at 1193–1303 K along with strain rates of 3 × 10^−4^–10^−3^ s^−1^ and dynamic recrystallization (DRX) or globularization of lamellar α phase is dominant mechanism of microstructure evolution. Zhou et al. [17] studied hot workability of near-α titanium alloy Ti–6Al–3Nb–2Zr–1Mo with an initial duplex microstructure by means of isothermal compressions. By integrating process maps and constitutive relationship, they proposed that the optimal hot working domain appeared at condition of 1198-1248 K/0.01–0.1 s^−1^, and the main mechanism of microstructure evolution in α + β phase field is dynamic recrystallization of β phase and super-plasticity deformation (SPD). Su et al. [18] have study the hot deformation behavior of near-α DsTi700 titanium alloy and revealed that the deformation mechanisms are mainly kinking of α lamellas or flow localization at high strain rate of 0.5–1 s^−1^ in α + β two phase field, while DRV is the dominated deformation mechanism in β single phase. Morakabati and Hajari [19] studied the high temperature deformation behavior of the Ti−5.7Al−2.1Sn−3.9Zr−2Mo−0.1Si (Ti-6242S) alloy with an initial acicular microstructure by using processing map and revealed that the dynamic globularization occurred in lower temperature conditions of α/β two-phase region (T ≤ 1223 K) and DRV was observed at high temperature (T = 1273 K). All above studies have been focused on as-cast alloy. However, the study of hot deformation behavior for PM alloy is rarely reported at present. Ding et al. [20] investigated the PM Ti600 alloy prepared by a HDH titanium powder-based PM-extrusion-annealing method and revealed that the microstructure was predominant by equiaxed grains after annealing below *β* transition temperature, while the bi-model structure was dominant after annealing at 1323 K in the β single phase. In our previous study [21], hot deformation of TiH_2_-based PM Ti-1100 at a relatively low temperature range of 973–1173 K and strain rate of 0.01–1 s^−1^ was investigated. The results of strain rate sensitivity and temperature sensitivity analysis showed that the flow instability mainly occurred at strain rates of 0.01 and 1 s^−1^ under all temperature range and the safe working window was limited at a narrow zone of 1073–1173 K with strain rate of 0.1 s^−1^. 

This paper mainly focused on the hot deformation behavior of TiH_2_-based PM near-α titanium alloy Ti-1100 in the upper α or α/β two phase region of 1123–1323 K and strain rates of 0.01–1.0 s^−1^. For the characterization of hot deformation behavior of Ti-1100 alloy, the strain hardening exponents, strain rate sensitivity, microstructure, and processing map were investigated.

## 2. Materials and Methods

Titanium hydride powder (Ti-4.5H-0.12O-0.015C-0.023N-0.016Fe in mass%) with a particle size of 45 μm and high grade of Al, Sn, Zr, Mo, and Si powders with a particle size of 75 μm were well mixed by the ball-mill machine. The mixture was subjected to cold isostatic pressing (CIP) at molding pressure of 200 MPa and holding time of 180 s to obtain the cylindrical lump alloy, and then the vacuum sintering was conducted at 1423 K for 240 min. under Ar atmosphere of 80~120 Pa. Finally, the CIP sintered lump was subjected to hot isostatic pressing (HIP) at 1223 K and 200 MPa for 180 min. 

In the hot compression test, a cylindrical sample with dimensions of φ 8 mm × 12 mm (in accordance to ASTM E209, preserving the aspect ratio) was machined along the axis of the HIP cylinder with a wire-electrode cutting machine. The hot compression tests of 15 samples were conducted by THERMECMASTER–Zthermos–simulation machine at deformation temperatures ranging from 1123 K to 1323 K with an interval of 50 K and strain rates of 0.01, 0.1, and 1 s^−1^. The whole isothermal compression test process was carried out under a vacuum degree lower than 10 Pa to prevent the samples oxidation. The specimens were heated to the target temperature at a rate of 10 K/s and held for 180 s. The fixed height of each sample was reduced by 60%, and the samples were quickly cooled to room temperature in order to retain high temperature deformation microstructure. To minimize friction, 0.1 mm thick mica wafer was placed at both end faces between the specimens and die. The true stress–strain curves were acquired automatically by a computer-equipped monitor. 

The isothermal compression specimen was cut along the radial direction, and the cutting section of specimen was grinded with SiC paper followed by mechanical polishing with a solution of water and diamond polishing paste. Then, the specimens were subjected to the etching in Kroll’s reagent (1 mL HF, 5 mL HNO_3_, and 100 mL water). Lastly, the microstructures of the central area of the cut specimens were characterized by optical microscopy (OM, DMM-490C, Caikang Optical Instrument Co., LTD, Shanghai, China) and scanning electron microscope (SEM, TESCAN-MIRA 4, Tescan Co., LTD, Shanghai, China branch).

The microstructure of the as-sintered alloys is shown in Figure 1. It shows a typical two-phase microstructure, mainly consisting of lath α phase with a width of about 20 μm, and β phase in the α phase boundary. From preliminary density measurements, the density of as-sintered alloy was determined to be 97.1% of the theoretical density, and the chemical composition of sintered alloy is shown in Table 1. The *β*-transus temperature of the alloy was preliminarily determined as 1293 K by differential scanning calorimeter (DSC), and the details of DSC analysis can be found in Appendix A.

## 3. Results and Discussion

### 3.1. True Stress–Strain Curves

#### 3.1.1. General Behavior 

Shown in Figure 2 is the macro-appearance of deformed samples at different processing conditions. It can be seen from the top and side views in Figure 2 that severe cracks were observed in the specimens deformed at 1123 K/0.1 s^−1^ and 1173 K/1 s^−1^, indicating that the sample under these conditions may not be suitable for thermal deformation. Likewise, slight cracks were observed on the side of the drum-shaped samples deformed at high temperature of 1273–1323 K and strain rate of 0.01 s^−1^. It should be noted that slight cracks on the side of the specimen are usually caused by compressive stress and additional tensile stress, which is different from the deformation behavior of the central part of the specimen subjected to a single pressure, so the hot workability in this case cannot be judged by these macroscopic appearances alone [22]. Under other experimental conditions, there were no obvious macro-cracks and micro-cracks, and the deformation of the sample is uniform.

Based on the hot compression experimental data of Ti-1100 alloy, the flow stress–strain curve was obtained, as shown in Figure 3. As seen at the beginning of the hot deformation process, the flow stress increases rapidly to the peak stress (σ_P_). This hardening effect is mainly attributed to the dislocation proliferation and intersection [23].

Beyond the peak stress, the flow stress exhibits the different behaviors. At a strain rate of 0.01 s^−1^, the flow stress exhibits softening response under all deformation temperatures, and this softening effect is more obvious at low temperature range. At a strain rate of 0.1 s^−1^, the flow stress exhibits the general behaviors of softening response in the low part of *α*/*β* region (T ≤ 1223 K). By comparison, the flow curves reached to steady-state condition at upper part of *α*/*β* region (T = 1323 K). This behavior can be explained by the fact that at a high temperature of 1323 K, the volume fraction of *β*-BCC phase with low mechanical property and high diffusivity is increased [24]. As the strain rate increases to 1 s^−1^, the flow stress curves show increasing tendency in the high temperature range of 1273–1323 K, implying work hardening effect was dominant at this condition. Obviously, these hardening effects are more significant at high strain rate (1 s^−1^) compared to that at 1273 K and 0.1 s^−1^. These behaviors indicated that dynamic softening by temperature and the work-hardening by strain rate interact each other, resulting the different behaviors under the different temperature and different strain rate conditions. As consequence, it shows the softening effects at low strain rate of 0.01 s^−1^, while both softening and hardening effects were observed at relatively high strain rate of 0.1–1 s^−1^ depending on temperature and strain rate.

#### 3.1.2. Effect of Temperature on Flow Stress

The true stress of the alloy is sensitive to the deformation temperature and strain rate. Figure 4 shows a typical example of temperature dependence on flow stress at a strain of 0.1 under the different strain rates. As seen in Figure 4, at constant strain rate, the true flow stress decreases continuously with the increase of temperature. Obviously, the decreasing rate of the flow stress decreases with increasing temperature, and this behavior can be attributed to the increasing the volume fraction of *β* phase with higher diffusivity and high stacking fault energy against to *α* phase [20,24]. Another possible cause of temperature dependence of flow stress is the high kinetics of *β* to *α* transformation. Thus, as *β* phase has lower strength and higher active slip system than *α* phase [25], the true flow stress decreases by an increase in deformation temperature. Similar behavior can be found in the deformation of IMI834 alloy [26] and Ti-6242S alloy [19].

### 3.2. Strain Hardening Exponents

To analyze the characteristics of hot work hardening of alloy, the concept of strain hardening was introduced based on the following functional model [27]:(1)$$\sigma =\sigma_{0}+K\varepsilon^{n_{}}$$
where $\sigma_{0}$ is the yield strength (MPa); K is the strength coefficient; ε is the strain; *n* is the strain hardening exponent, which can be express by Equation (1) [28]:(2)$$n={\left[\frac{\partial \left(lg\sigma \right)}{\partial \left(lg\varepsilon \right)}\right]}_{\;\dot{\varepsilon },\;T}$$

The parameter *n* reflects the comprehensive effect of strain hardening and flow stress softening process [28,29]. It can be known from Equation (2) that the negative value of *n* represents the trend of stress softening, while the positive value of *n* represents the work hardening behavior. By the data processing and substitution into Equation (2), the strain hardening exponent *n* was obtained. Figure 5 shows the influence of temperature on the strain hardening exponent *n* under different strains. At a relatively low strain rate of 0.01 s^−1^, the values of *n* under a strain of 0.1 shown in Figure 5a fluctuates near the balance state line of *n* = 0 with temperature. However, the value of *n* at a strain rate of 0.01 s^−1^ are mostly negative as the strain rate increases to 0.3 and 0.5, and it becomes more negative at higher temperature. When the strain rate is 0.1 s^−1^ and 1 s^−1^, the values of *n* shown in Figure 5b,c are all negative in the temperature range of 1123–1223 K, and it becomes positive at temperature above 1223 K, meaning that the softening effect has been changed to the hardening effect near at 1223 K. Where the temperature is 1323 K and strain rate is 1 s^−1^, the value of *n* increases from the minimum value of −0.8 to its maximum value of 0.8. Since the strain hardening exponent *n* is a parameter of comprehensive effects of strain hardening and flow stress caused by thermal effects, the increase of the *n* value is basically caused by the decreases of softening effect as the deformation temperature increases [28]. Similar behavior of the increasing tendency with the temperature was reported at SP700 superplastic titanium alloy under the high strain rate of 0.1 s^−1^ [30].

Figure 6 shows the variation of the strain hardening exponent of the alloy with the change of strain under the different strain rates. It can be found that at temperatures of 1123 K under all strain rates, the curve of *n* value generally decreases during strain of 0.6 and approaches to stable state, indicating that the flow softening was gradually increased due to the temperature rise effect, and then decreased as temperature rise effect disappears at the high strain [30]. Moreover, under all strain conditions, the value of *n* decreases more negatively with the increase of strain rate. Similarly, at temperature of 1173 K, the *n*-value curve under all strain conditions moves to a more negative region with the increase of strain rate. As seen in Figure 6a at the temperature of 1273 and 1323 K, the thermal softening effect is more significant when the strain rate is 0.01 s^−1^. However, when the strain rate increases to 0.1 s^−1^, the strain hardening effect is dominant, and it becomes more significant as the strain rate further increases to 1 s^−1^ as shown in Figure 6c. It was revealed from Figure 6b,c that, the higher temperature the more prone to strain hardening and the lower temperature the more prone to thermal softening. At the temperature of 1223 K, the value of *n* is negative under all strain conditions and strain rates, and slightly increases with increase of strain rate. No significant change of *n* value with strain was observed at this temperature, showing better uniform plastic deformation ability.

It is noted that the softening effect in term of negative *n*-value only considers the temperature rise effect, but not the microstructural evolution related softening effects such as dynamic recrystallization (DRX) or dynamic recovery (DRV). In general, the softening caused by DRX or DRV is favorable to hot deformation. However, the softening caused by temperature rise is not conducive to hot deformation. It is generally known that the temperature rise during alloy deformation will generate deformation heat, leading to instability phenomena such as local flow or adiabatic shear [31]. In this work, the most severe flow softening with minimum value of *n* occurred at low temperature of 1123 K with a high strain rate of 1 s^−1^, so the deformation in this condition may be dangerous and should be avoided. In addition, obvious flow softening effect were also observed at high temperature of 1323 K with low strain rate of 0.01 s^−1^. This is mainly due to the formation of the *β* phase as discussed in detail in the following section.

### 3.3. Microstructure Observation

As seen in Figure 3, at low temperature range of *α*/*β* region (T ≤ 1273K), the strain–stress curves show a peak stress at relatively low strain followed by flow softening, while flow curves show steady-state behavior at high temperature (T = 1323 K). In single-phase alloys, these two behaviors are often attributed to occurrence of DRV, i.e., the occurrence of dislocation climb and DRX, i.e., the nucleation and growth of new grains, respectively. However, different microstructural evolution is known to be responsible for the observed flow softening in two-phase Ti-alloys [32]. 

The microstructures of Ti-1100 alloy compressed at 1123 K, 1273 K, and 1323 K under the strain rates of 1~0.01 s^−1^ are presented in Figure 7(a1–a3), (b1–b3), and (c1–c3), respectively. According to Figure 7, the evolution of randomly distributed *α* lamellar phase with a length of 10~20 µm and a thickness less than 10 µm, during isothermal deformation, was a prevailing microstructure feature at deformation temperature below 1273 K, while at 1323 K, the various form of transformed *β* phase with non-lamellar *α* structure is its main feature, mainly due to an increase in volume fraction of *β*-BCC phase. During deformation at 1123–1273 K, the *α* colony, characterized by the fine parallel array structure, was observed, and tends to rotate perpendicular to the compressive axis. This, *α* colony formation is caused by local overheating. Due to the poor thermal conductivity of Ti-1100 alloy, the distortion energy caused by deformation cannot be released in the form of heat immediately, which may further lead to local overheating and the formation of fine parallel structure [33]. There is no specific grain size change with the change of strain rate. In addition, the bending/kinked lamella and sub-grain formation are the general characteristics of microstructural evolution. It is generally known that the thick initial lamellar microstructure undergoes buckling and elongates, while thin microstructure is more easily fragmented, so the bending/kinked lamella are possibly formed under the different deformation conditions. The sub-grain formation may be caused by boundary-splitting mechanism through intense localized shear during hot deformation [20]. Moreover, the globular shape or equiaxed grains with a size less than 5 µm are usually observed near the *α* lath boundary, indicating that dynamic recrystallization (DRX) of *α* phase occurs. This is because the fractured lamellar grains during hot deformation promote the formation of refined equiaxed microstructure through recrystallization of α grains. [4]. 

Differently, as observed in Figure 7(c1–c3), the microstructure of the alloy deformed at 1323 K has the characteristics of transformed *β* phase with different structures. As seen, the microstructure of the specimen deformed at the strain rate of 1 s^−1^ contained transformed *β* phase with basketweave (widmanstätten) structure, and discontinuous α layers at the transformed *β* grain boundary, and also within grain. This morphorlogy of the transformed microstructure is usually formed near *β*-transus temperature [34]. Clearly, compared to lower temperature, the volume of *β* phase at this condition was increased and the boundary of *α* phase becomes more irregular. As the strain rate decreases to 0.1 s^−1^, the proportion of transformed *β* increased and the α/*β* grain boundary became blurred. When the strain rate is further decreased to 0.01 s^−1^, the enrichment of transformed *β* near the *α* phase boundary was observed. In addition, the globularized *α* phase was also observed. In this case, the globularization of *α* phase is mainly controlled by termination migration mechanism. It is believed that the *β* phase diffuses into the boundary as an interlayer phase, so it can reduce the dihedral angle and further leads to grooves in the α-layers. In other words, the mass transfer from the curved surfaces of the lamellar grain terminations to the vicinal flat surface of the lamellae drives the termination migration [35]. More clear observations and phase identifications of this condition can be seen from the SEM micrographs in Figure 8. Figure 8a,c shows the secondary electron image (SE) of Ti-1100 alloy deformed at 1323 K/ 0.01 s^−1^ and 1323 K/1 s^−1^, respectively, in which both α and *β* phases can be easily recognized. Figure 8b presents back-scattering electron (BSE) image of alloy deformed at 1323 K/1 s^−1^. As shown by the white arrow in Figure 8b, due to the shear strain applied during deformation, there exists sub-boundary in “A” layer. As an unstable interface, this sub-boundary provides favorable conditions for *β* phase penetration. As a consequence, the α lamella is fractured and the *α*/*β* interface is formed. Similar observations have been reported previously for globularization of Ti-6242S alloy [20]. For phase identification, EDS (energy dispersive spectroscopy) analysis was conducted for matrix *α* phase (point 1 in Figure 8b) and transformed *β* phase (point 2 in Figure 8b), and the results are shown in Figure 8d. It can be seen from the EDS analysis that the α-stable element of Al is higher in the matrix, while the *β*-stable element of Mo or Si is rich in phase with fine parallel structure, thus confirming that the matrix is α phase and the other is the transformed *β* phase.

In general, DRX or DRV is the main mechanism of the flow stress softening in hot deformation. Usually, DRX occurs more obviously in the sample deformed at lower temperature and lower strain rate [22]. In present study, the flow stress softening is mainly controlled by DRX and microstructural evolution. As seen in Figure 7, dynamic recrystallization occurred in the alloys deformed in lower part of two-phase *α*/*β* region (T ≤ 1273 K), while microstructural evolution of transformed *β* grain and dynamic globalization of *α* by diffusional control are the dominant mechanism of flow softening at high temperature of 1323 K.

### 3.4. Strain Rate Sensitivity

It is widely known that the strain rate sensitivity (*m*) is a parameter of great importance to determine the deformation behavior of the material. The high strain rate sensitivity can either induce the occurrence work hardening or flow stress softening depending on the temperature and strain [21]. In general, the positive *m* value is favorable to the plasticity of the material, while the negative *m* value is adverse to hot deformation because the material is prone to the defects [30]. In addition, when the *m* value is greater than 0.5, the superplasticity occurs [36].

In terms of the high temperature deformation theory, the strain rate sensitivity is given in the following [37]:(3)$$m={\left[\frac{\partial ln\sigma }{\partial ln\dot{\varepsilon }}\right]}_{T,\;\varepsilon }\approx {\left[\frac{\Delta ln\sigma }{\Delta ln\dot{\varepsilon }}\right]}_{T,\;\varepsilon }$$

After data processing, the values of *m* were obtained. Figure 9 shows the variation of strain rate sensitivity under different deformation conditions. As seen in Figure 9a at the strain rate of 0.01 s^−1^, the *m* values are all positive and greater than 0.18. It generally shows increasing tendency with the increase of strain under most of temperature conditions, indicating the plasticity of material is enhanced as the strain increases. When the strain rate is increased to 0.1 s^−1^ and 1 s^−1^, the strain rate sensitivity behaves differently depending on deformation temperature. As seen in Figure 9b,c under low temperature of 1123–1173 K, the *m* value decreases with the increase of strain, and this reduction of *m* is more significant at higher strain rates. These phenomena implied that the deformation mechanism has been changed as increasing the strain rate. When the high strain rate is 1s^−1^, the value of *m* becomes negative as the strain approaches above 0.3, showing poor plasticity of material at this domain.

In all cases, the peak zone of *m* value mostly appeared in the high temperature range of 1273–1323 K and the strain of 0.6, indicating good plasticity of material. The maximum *m*-value of 0.64 was obtained at 1273 K and strain rate of 0.6, and the maximum *m*-values of 0.52 and 0.56 under strain rate of 0.1 s^−1^ and 1 s^−1^ were obtained at 1273 K and strain rate of 0.6. These values are all greater than 0.5, indicating the material has superplasticity. This is mainly due to the increase in volume fraction of *β* phase transformed near α/β temperature region, resulting in higher *m* value and easy deformation. According to superplastic theory, the strain rate sensitivity value can be affected by the deformation parameters such as T, ε, and $\dot{\varepsilon }$, alloy composition, and phase fraction. Earlier studies on Ti60 [38], and Ti-6242S [20] alloys have shown that the maximum *m*-value was obtained at 0.93 *T_β_*. Similarly, in the present study, the maximum *m*-value was obtained in the temperature range of 1273–1323 K, near the *β*-transus temperature.

The strain rate sensitivity near 1223 K is relatively stable with strain under a fixed strain rate, but it increases with the increase of strain rate. When the strain rate is 0.01 s^−1^ and 0.1 s^−1^, the *m* values are in the range of 0.18-0.30, which is a typical deformation controlled by climb-limited glide processes characteristic of power-low creep or dislocation glide [34]. When the strain rate is 1 s^−1^, the *m* values are higher than 0.3, which is probably related to grain-boundary sliding (GBS) [28].

### 3.5. DMM Processing Map

In order to characterize the hot working behavior of materials under plastic deformation without losing stability at high temperature, processing map of a material based on the dynamic materials model (DMM) [39] was developed. This theory assumes that the workpiece is regarded as power dissipation in high temperature deformation and the power observed by workpiece essentially dissipates during the hot deformation process. In this map, the microstructure evolution such as DRV, DRX, and superplasticity are considered as “safe” deformation conditions, while the “unsafe” deformation conditions are various defects such as shear band and flow localization. To develop the processing map of the alloy, the efficiency of power dissipation (*η*) representing the power dissipation capacity is defined as Equation (4):(4)$$\eta \;\left(T,\;\dot{\varepsilon }\;\right)=\frac{2m}{m+1}$$
where *m* is the strain rate sensitivity coefficient for a given flow stress and expressed by *m* = $\partial \left(\ln\sigma \right)/\partial \left(\ln\dot{\varepsilon }\right)$. Variation of power dissipater efficiency with temperature and strain rate constitutes the power dissipater map, which presents various power domains correlated to specific microstructural mechanisms.

To identify the microstructural instability, Ziegler flow instability [40] was used as Equation (5):(5)$$\xi \left(\dot{\varepsilon }\right)=\frac{\partial \ln\left[m/\left(m+1\right)\right]}{\partial ln\dot{\varepsilon }}+m$$
where $\xi \left(\dot{\varepsilon }\right)$ is a dimensionless instability parameter. This model assumes that when the instability parameter is negative, the flow instability of material occurs. The variation of instability criteria with strain rates and temperature constitutes the instability map. By superimposing power dissipater map and instability map, the processing map of a material is developed. Figure 10a,b show the processing maps of the alloy at the strain of 0.3 and 0.6, respectively. As seen, the value of *η* generally increased with increasing temperature, and there are multiple peak domains in high strain rate region and low strain region. As shown in Figure 10a, the instability mainly occurred in the area of temperature below 1173 K and higher strain rate, and this instability areas coincided with low efficiency regions. This instability region gradually disappears as the temperature increases and the strain rate decreases. This reduction of instability may be attributed to the reduction of deformation heating effects, which is considered as main factor of an adiabatic shear banding [31]. With the increase of strain, the instability regions expanded to the temperature of 1173 K and to the strain rate of 0.023 s^−1^ (ln$\dot{\varepsilon }$ = −3.78) as show in Figure 10b. In addition, another instability zone was appeared at a high strain rate of 1s^−1^ and a temperature of 1273 K. The optimum processing range of material deformation is 1273–1323 K along with strain rate range of 0.01–0.165 s^−1^ (ln$\dot{\varepsilon }$ = −4.6~−1.8) since the power dissipation efficiency generally shows high values at this region and the instability criteria has a positive value.

Figure 11 shows the microstructures of specimens deformed under various deformation conditions. It can be known from microstructural observation that the shear deformation is the dominant instability mechanism. As seen in Figure 11a,b, the microstructure exhibited a severe flow localization at near 45° to the compression axis and large amount of shear band. In general, the localized plastic flow or shear deformation are easy to form at low temperatures and high strain rates [41], and similar results have been observed in Ti-6242S alloy [19]. As described in Section 3.2, the minimum value of strain hardening exponents *n* at 1123 K also demonstrated severe flow softening due to temperature rise. Another instability is inhomogenous microstructure as shown in Figure 11c at 1273 K/1 s^−1^. This may be due to the insufficient phase transformation behavior occurring at high strain rate and high temperature, which leads to the inhomogeneity of microstructure. These microstructures are undesirable in achieving appropriate mechanical properties, so it should be avoided during hot deformation.

According to DMM, *η* is related to the microstructural mechanism which occurs dynamically to dissipate power, such as DRV, phase transformation, and grain refinement [42]. The higher *η* leads to a better dissipative behavior from the workpiece associated with microstructure evolution, resulting better deformation capacity. On the other hand, the lower *η* leads to worse dissipative behavior due to temperature rise effect. Therefore, it was worth to investigate the variations of *η* under different processing conditions. Shown in Figure 12a is the variation of *η* with strain at different temperatures. At 1123 K shown in Figure 12a, the values of *η* are all lower than 0.5, and it decreased more with the increase of strain. This behavior implies that the deformation power was mainly dissipated into viscoplastic heat due to temperature rise effect, indicating low formability at warm deformation region. So, this is the reason why the alloy is prone to defects when it is deformed at low temperature as discussed above. As shown in Figure 12a at moderate temperature of 1223 K, the value of *η* remains constant with strain, but it shows increasing tendency with strain at higher temperatures of 1273 K and 1323 K. Obviously, the slope of *η* with respect to strain gradually increases with increase of temperature, and this means that the power dissipation mechanism during hot deformation has been changed from temperature-dependent to microstructure-dependent with the increase of temperature. The variations of power dissipation efficiency with temperature under different strains and a fixed strain rate of 0.1 s^−1^ is shown in Figure 12b. Clearly, the power dissipation efficiencies increased with increasing temperature and the increase in *η* values is more significant with increase of strain. As seen in Figure 12a,b at fixed strain of 0.6, the value of *η* increases with increase of temperature, implying microstructure optimization were more likely to occur at higher temperature. These findings exactly explain the different temperature-dependent behavior of true stress–strain curves as discussed in preceding sections. Figure 12c shows the variations of power dissipation efficiency with strain under different strain rates and the fixed temperature of 1223 K. The *η* values remained constant with the strain at each strain rate. The *η* values at a strain rate of 0.01 s^−1^ remained about 0.6, implying microstructural changes such as DRX or DRV are more likely to occur in this case. Metallographic observation under this condition shown in Figure 7 also confirmed the DRXed microstructures. On the other hand, the *η* values at a strain rate of 1 s^−1^ are in the range of 0.3–0.4 and lower than 0.5, implying the microstructural optimization is difficult to occur in this case, but mostly consumed by viscoplastic heat or local overheating which may induce the instability such as shear deformation or localized plastic flow.

## 4. Conclusions

The true stress–strain curves showed that at a strain rate of 0.01 s^−1^, the flow stress exhibited softening response under all deformation temperatures. At relatively high strain rate of 0.1 s^−1^ and 1 s^−1^, the flow stress exhibits the general behaviors of softening response in the low part of *α*/*β* region (T ≤ 1223 K), while both softening and hardening effects were observed at upper part (T > 1273 K).The strain hardening exponent (*n*) analysis revealed that the hardening/softening effects become more significant as the strain increases. In addition, the higher temperature the more prone to strain hardening and the lower temperature the more prone to thermal softening. When the strain rate is 0.1 s^−1^ and 1 s^−1^, the values of *n* are all negative in the temperature range of 1123–1223 K and it becomes positive at temperature above 1223 K, implying that the softening effect has been changed to the hardening effect near at 1223 K. The most severe flow softening with minimum value of *n* occurred at low temperature of 1123 K with high strain rate of 1 s^−1^, and high temperature of 1323 K with low strain rate of 0.01 s^−1^.Lamellar phase with a length of 10~20 µm and a thickness less than 10 µm was the main structure of alloy deformed at temperature below 1273 K, and the various form of transformed *β* phase with irregular shape of *α* structure is the main feature at temperature of 1323 K. The dynamic recrystallization of *α is* the main softening mechanism in lower part of temperature (T ≤ 1273 K), while microstructural evolution of transformed *β* grain and dynamic globalization of *α* by diffusional mainly control the softening effect of alloy.The developed processing map demonstrated that the deformation in the temperature range of 1273–1323 K and strain rates of 0.01–0.165 s^−1^ was desirable and led to high efficiencies. At low temperature of 1123 K, increasing strain rate led to the increase of flow instability, which was primarily manifested as localized plastic flow, adiabatic shear bands and inhomogenous microstructure. The variation of power dissipation energy (*η*) with strain demonstrated that the power dissipation mechanism during hot deformation has been changed from temperature-dependent to microstructure-dependent with the increase of temperature for the alloy deformed at 0.1 s^−1^.

## Figures and Tables

**Figure 1 materials-15-05932-f001:**
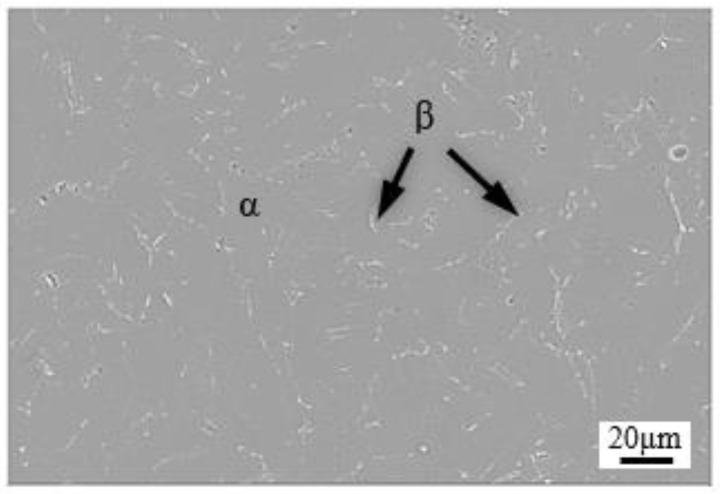
Microstructure of as-prepared Ti-1100 alloy.

**Figure 2 materials-15-05932-f002:**
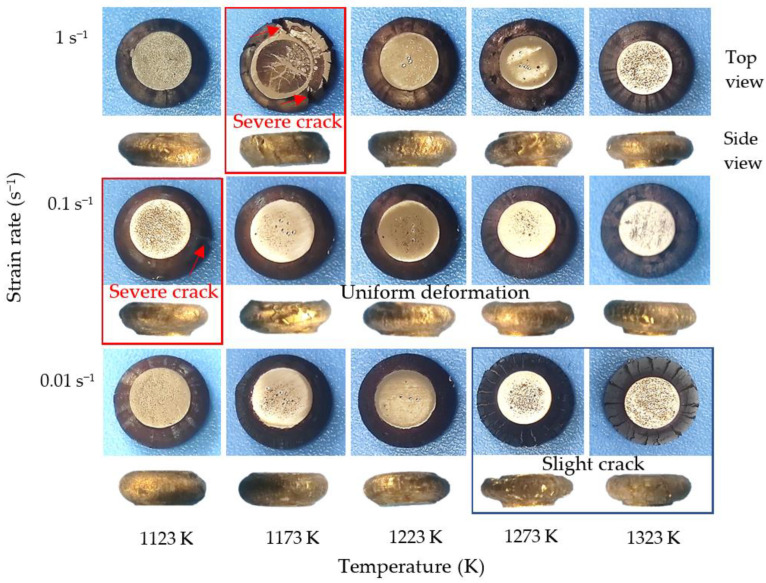
Macro-appearance (top and side views) of deformed alloys at different processing conditions.

**Figure 3 materials-15-05932-f003:**
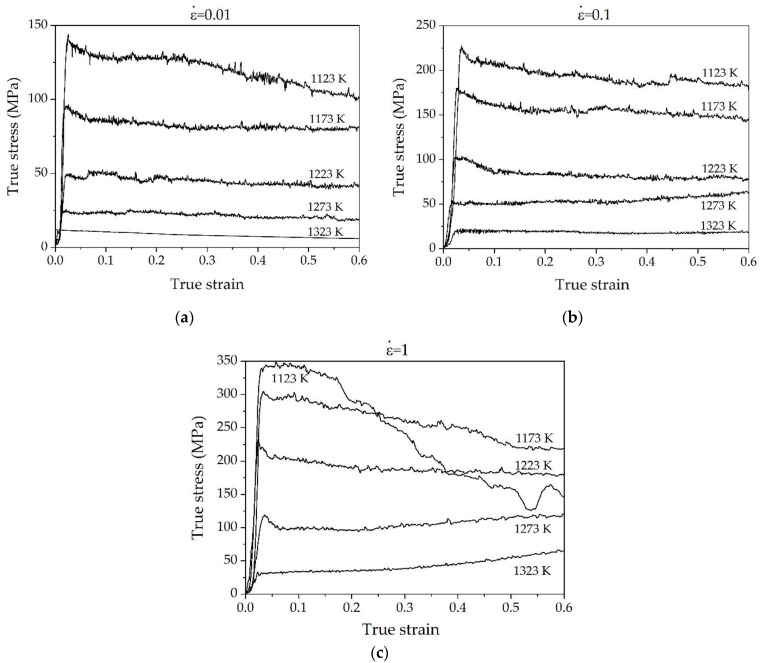
The true strain–stress curves of Ti-1100 alloy at different deformation conditions: (**a**) 0.01 s^−1^, (**b**) 0.1 s^−1^, and (**c**) 1 s^−1^.

**Figure 4 materials-15-05932-f004:**
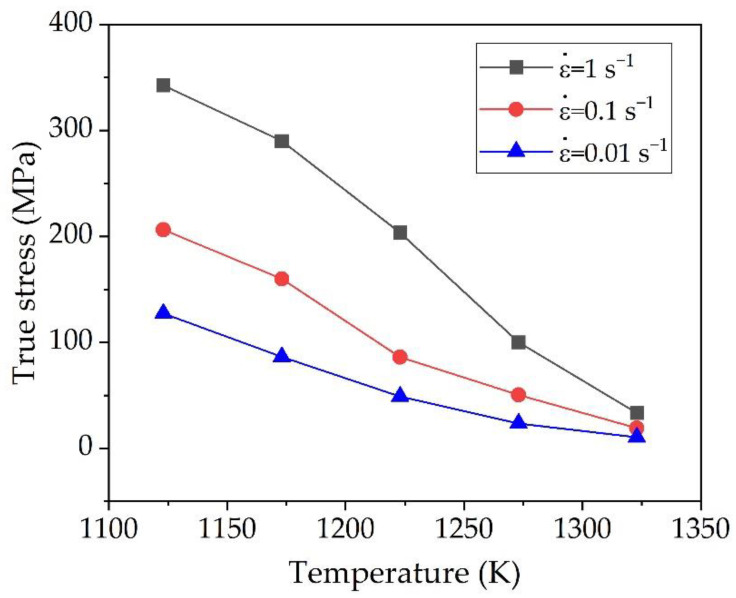
Temperature dependence of the flow stress at a strain of 0.1 under various strain rates.

**Figure 5 materials-15-05932-f005:**
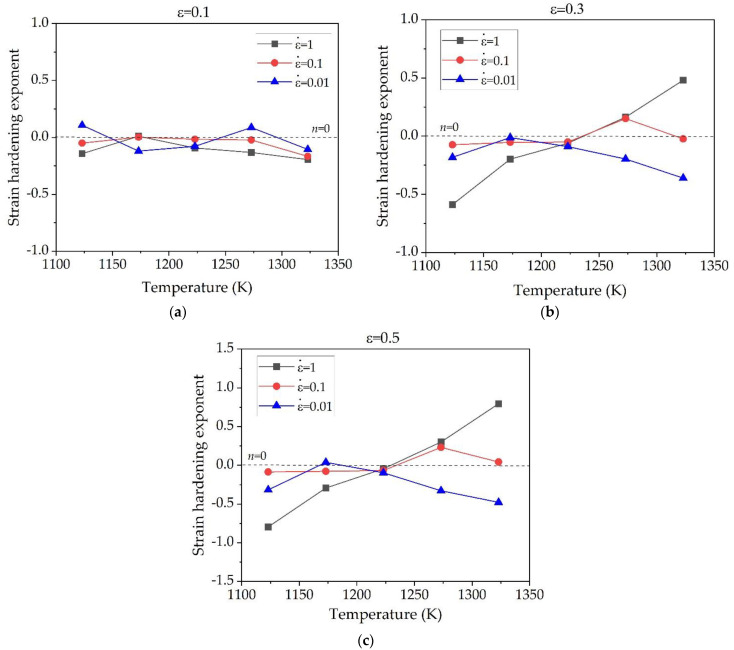
Influence of deformation temperature on the strain hardening exponent *n* of Ti-1100 titanium alloy under different strains: (**a**) ε = 0.1, (**b**) ε = 0.3, and (**c**) ε = 0.5.

**Figure 6 materials-15-05932-f006:**
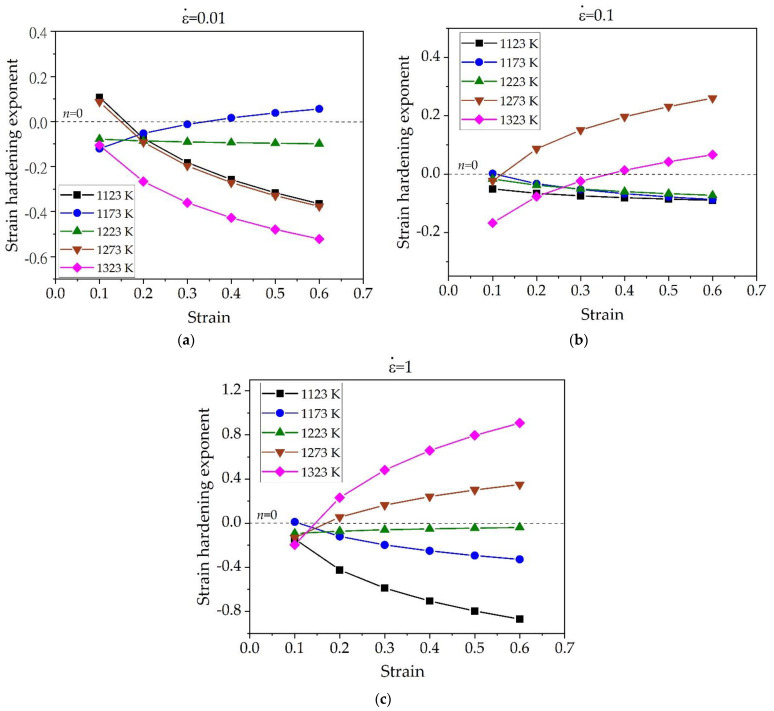
Variation of the strain hardening exponent as a function of strain under different strain rates: (**a**) $\dot{\varepsilon }$ = 0.01 s^−1^, (**b**) $\dot{\varepsilon }$ = 0.1 s^−1^, and (**c**) $\dot{\varepsilon }$ = 1 s^−1^.

**Figure 7 materials-15-05932-f007:**
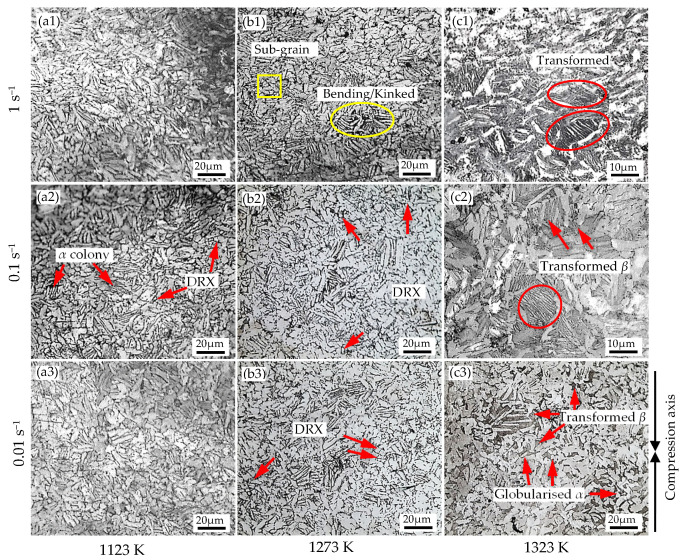
Microstructure of Ti-1100 alloy with a strain of 0.6 at (**a1**) 1123 K/1 s^−1^, (**a2**) 1123 K/0.1 s^−1^, (**a3**) 1123 K/0.01 s^−1^; (**b1**) 1273 K/1 s^−1^, (**b2**) 1273 K/0.1 s^−1^, (**b3**) 1273 K/0.01 s^−1^; and (**c1**) 1323 K/1 s^−1^, (**c2**) 1323 K/0.1 s^−1^, (**c3**) 1323 K/0.01 s^−1^.

**Figure 8 materials-15-05932-f008:**
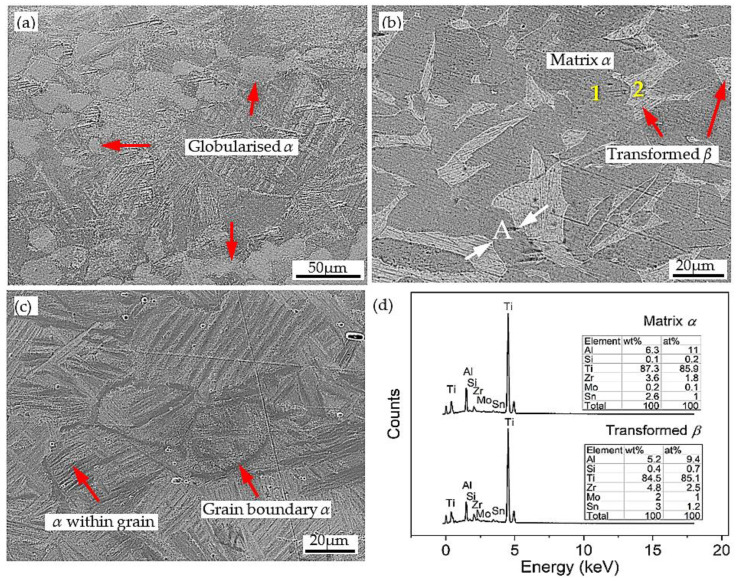
SEM micrographs of Ti-1100 alloy compressed at 1323 K: (**a**) SE image at 0.01 s^−1^, (**b**) BSE image at 0.1 s^−1^, (**c**) SE image at 1 s^−1^, and (**d**) EDS analysis of point 1 and 2 in Figure 8b.

**Figure 9 materials-15-05932-f009:**
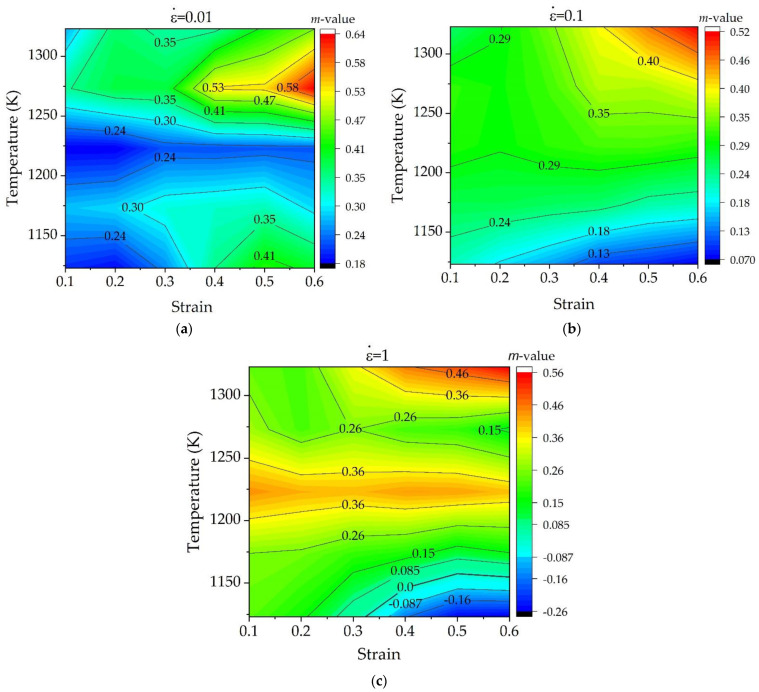
Contour maps of strain rate sensitivity under different strain rates: (**a**) $\dot{\varepsilon }$ = 0.01 s^−1^, (**b**) $\dot{\varepsilon }\;$= 0.1 s^−1^, and (**c**) $\dot{\varepsilon \;}$ = 1 s^−1^.

**Figure 10 materials-15-05932-f010:**
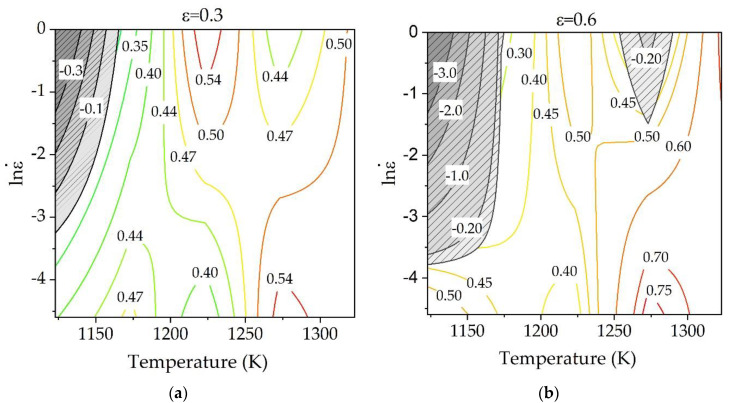
Processing maps of Ti-1100 alloy at strains of (**a**) 0.3 and (**b**) 0.6 (shaded zone represents instability domain; numbers represent power dissipation efficiency (%)).

**Figure 11 materials-15-05932-f011:**
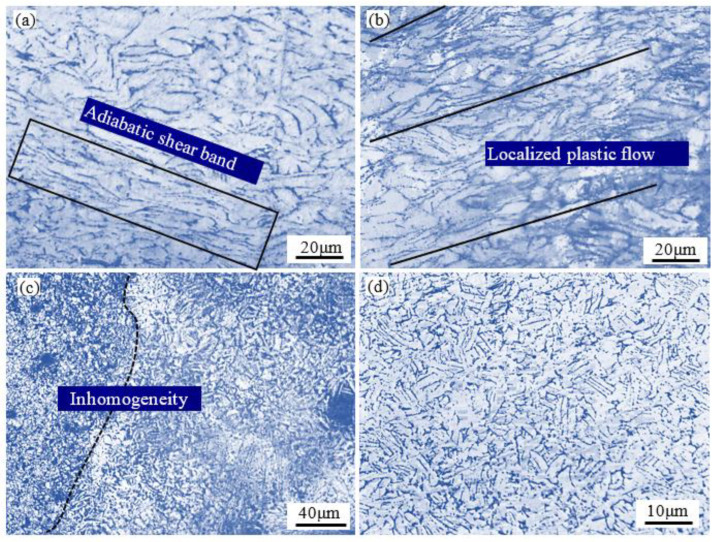
Optical micrographs of specimens deformed under different conditions: (**a**) 1123 K, 1 s^−1^, (**b**) 1123 K, 0.1 s^−1^, (**c**) 1273 K, 1 s^−1^, and (**d**) 1273 K, 0.1 s^−1^.

**Figure 12 materials-15-05932-f012:**
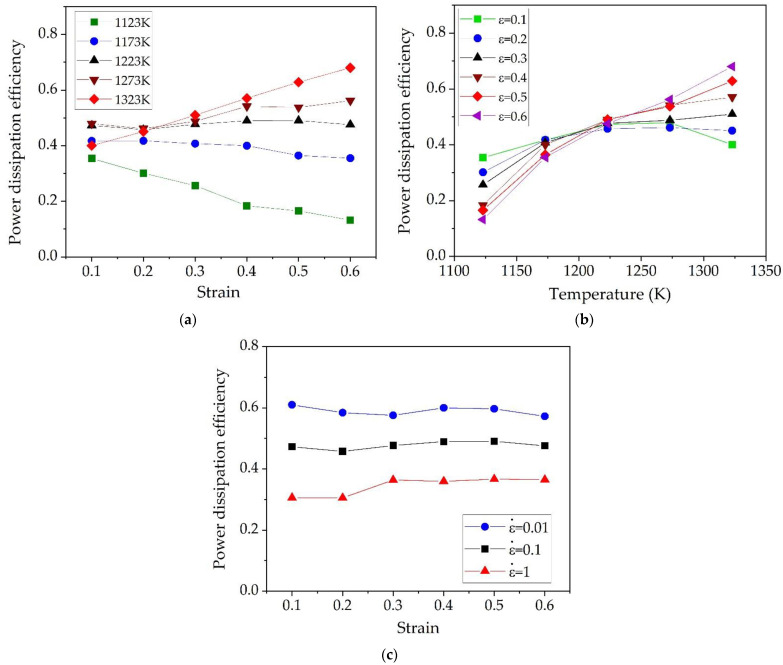
Variations of power dissipation efficiency with (**a**) strain at 0.1 s^−1^, (**b**) temperature at strain rate of 0.1 s^−1^, and (**c**) strain at 1223 K.

**Table 1 materials-15-05932-t001:** Chemical composition of sintered near-*α* Ti-1100 alloy (wt%).

Ti	Al	Zr	Sn	Mo	Si	C	N	H	O
88.3	5.05	3.689	1.96	0.32	0.29	0.0071	0.0013	0.0016	0.12

## Data Availability

Not applicable.

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
