# Peer review of "Characterization of Hot Deformation of near Alpha Titanium Alloy Prepared by TiH2-Based Powder Metallurgy"

_materials, 2022, doi:10.3390/ma15175932_

Round 1

Reviewer 1 Report

The presented paper is very interesting. 

The weakest part of the paper is not sufficient microstructural characterization. 

In Paragraph Materials and methods, no information about characterization methods was provided. 

Fig 7 and 8 are not very clear, lack of deeper phase analysis eg by EBSD method. EDS analysis are presented with two decimal places what is incorrect.  

Author Response

The authors appreciate very much for the Reviewer’s suggestions and comments which helped us to improve our manuscript. Those are very constructive and fruitful to enhance the quality of this manuscript. Please find our revised version of the manuscript. We have made some modifications according to the Reviewer’s suggestions and comments. Important modifications have been highlighted by blue color or red color in “track change” mode. A detailed description for each modification is presented in this letter.

  1. The weakest part of the paper is not sufficient microstructural characterization. 

A1. Supplementary notes have been provided in the revised manuscript.

In order to better understand the hot deformation behavior of alloy, it is usually necessary to analyze the microstructure of alloy, especially the phase evolution mechanism and microstructure defects of the alloy[15-20], so as to provide a basis for judging the optimal hot working conditions. In the Section 3.3 Microstructure observation, the phase evolution mechanism along with the general characteristics of microstructure such as phase identification, size, distribution, etc. were presented. It was concluded that the mechanism of phase evolution at deformation temperature of 1123K-1273K was mainly characterized by DRX of α-phase, while transformed β phase is the main characteristic of phase evolution of alloy at 1323K. The microstructure defects of the alloy were discussed on the DMM processing map in Section 3.5.

  1. In Paragraph Materials and methods, no information about characterization methods was provided. 

A2. Supplementary notes have been provided in the revised manuscript.

  The isothermal compression specimen was cut along the compression axis, and the cutting section of specimen was grinded with SiC paper followed by mechanical polishing with a solution of water and diamond polishing paste. Then, the samples were etched in Kroll’s reagent (composition: 1 ml HF, 5 ml HNO3, 100ml water). Lastly, the microstructures of the central area of cut specimens were characterized by optical microscope (DMM-490C, Shanghai Caikang Optical Instrument Co., LTD, Shanghai, China) and scanning electron microscope (SEM, TESCAN- MIRA 4, Tescan Co., LTD, Shanghai, China branch).

  1. Fig 7 and 8 are not very clear, lack of deeper phase analysis eg by EBSD method. EDS analysis are presented with two decimal places what is incorrect.  

A3. (1) In order to see Figure 7 and Figure 8 more clearly, image processing such as sharpening and contrast was performed on the revised manuscript.

Through general microstructure analysis, such as phase analysis by morphology and EDS, microstructure defects, etc., the mechanism of microstructure evolution can be possibly clarified and the thermal deformation behavior can be understood [15-21]. Therefore, EBSD analysis is not specifically carried out in this study. Similar works can be found in our previous works [Zhu, Y.L.; Yang, S.L, Ma, L.; Piao, R.X.; Iron Steel Vanadium Titanium, 2019, 40(5): 50 (in Chinese)]-[22], and also in the references of [ Long, S.; Xia, Y. F.; Hu, J. C. et al. Hot deformation behavior and microstructure evolution of Ti-6Cr-5Mo-5V-4Al alloy during hot compression. Vacuum, 2019, 160:171-180.] - [Liu, S.F.; Li, M. Q.; Luo, J. et al. Deformation behavior in the isothermal compression of Ti–5Al–5Mo–5V–1Cr–1Fe alloy. Materials Science & Engineering A, 2014, 589:15-22.].

(2) Fig.8(d) of EDS analysis with one decimal place has been updated.

Reviewer 2 Report

The results presented in the paper are interesting and well documented. The authors presented a wide range of research on both mechanical properties and microstructural observations. Due to the changes in the microstructure related to the phase transformation, it would be worth presenting the phase transformation temperature for this alloy (results of dilatometric tests). The work is an interesting contribution to a better understanding of changes taking place in titanium alloys during thermoplastic processing.

Author Response

The authors appreciate very much for the Reviewer’s suggestions and comments which helped us to improve our manuscript. Those are very constructive and fruitful to enhance the quality of this manuscript. Please find our revised version of the manuscript. We have made some modifications according to the Reviewer’s suggestions and comments. Important modifications have been highlighted by  blue color or red color in “track change” mode. A detailed description for each modification is presented in this letter.

The results presented in the paper are interesting and well documented. The authors presented a wide range of research on both mechanical properties and microstructural observations. Due to the changes in the microstructure related to the phase transformation, it would be worth presenting the phase transformation temperature for this alloy (results of dilatometric tests). The work is an interesting contribution to a better understanding of changes taking place in titanium alloys during thermoplastic processing.

  1. Thank you for your recognition of our work.

The β-transus temperature (Tβ, the temperature at which the α + β-phases transform to the β-phase) of the alloy was preliminarily determined as 1293K by differential scanning calorimeter (DSC) and it has been supplemented in the section 2 “Material and Methods”.

The raw data of DSC was attached in the Appendix of revised manuscript. As seen, Tβ temperature was determined as 1293K (or 1019.7°C).

Reviewer 3 Report

The aim of this study and the results obtained should be stated in the abstract, only the experimental parameters are insufficient. Does not fully reflect the study content

In line 53, it is stated that there is a limited study, but there are many studies in the literature on this subject.

Alpha is expressed with words in some places and symbols in some places, it should be rearranged.

In the Introduction, some studies that are not in the scope of the study or that are not similar are cited.

What are the other contions mentioned in line 156? This paragraph should be rewritten.

The results obtained in Figure.3 should be examined better, more literature studies can be presented.

Line 209-229 paragraph should be rewritten, the English of the expressions is insufficient.

In Figure.5a, the axes are wrong.

In Figure.7, the expressions in the lower part of the graph are insufficient. for example, what is the a1? a2? a3?

The interpretation of the data obtained in Figure.10 should be reconsidered. This is incomprehensible.

What is the difference between Figure 12b and 12c? What should be understood?

The conclusion should be expanded.

Author Response

The authors appreciate very much for the Reviewer’s suggestions and comments which helped us to improve our manuscript. Those are very constructive and fruitful to enhance the quality of this manuscript. Please find our revised version of the manuscript. We have made some modifications according to the Reviewer’s suggestions and comments. Important modifications have been highlighted by blue color. A detailed description for each modification is presented in this letter.

1.The aim of this study and the results obtained should be stated in the abstract, only the experimental parameters are insufficient. Does not fully reflect the study content

A1. Supplementary explanations for the main of this study and important finding of power dissipation energy mechanism change have been provided in the abstract of the revised manuscript.

2.In line 53, it is stated that there is a limited study, but there are many studies in the literature on this subject.

2A. In order to avoid unnecessary misunderstanding and consider the mentioned question 4, the sentences in line 52-53 have been deleted and simplified.

3.Alpha is expressed with words in some places and symbols in some places, it should be rearranged.

A3. The word “alpha” has been changed to α in the main content of revised manuscript.

4.In the Introduction, some studies that are not in the scope of the study or that are not similar are cited.

A4. Some part of the second paragraph of the introduction has been revised. This part mainly introduced the advantages of TiH2-based powder metallurgy method in the preparation and better performance of near-α titanium alloy. Please see the revised manuscript. Besides, the reference [12] which is not directly related to this study has been deleted.

5.What are the other contions mentioned in line 156? This paragraph should be rewritten.

A5. The Figure 2 has been updated and the relevant paragraph has been rewritten in the revised manuscript.

6.The results obtained in Figure.3 should be examined better, more literature studies can be presented.

A6. In this study, general behavior of true stress-strain curves was examined at peak stress first, and then the three different behaviors of flow stress curve beyond the peak stress were presented. Moreover, effect of temperature on flow stress was present. All important key features of stress-strain curve were explained, and appropriate reasons or explanation were given according to the relevant literature study. The detailed examination of three different behavior of flow stress curve were discussed in the following 3.2 section. Please see the section 3.2 and section 3.3.

7.Line 209-229 paragraph should be rewritten, the English of the expressions is insufficient.

A7. The mentioned paragraph has been rewritten in revised manuscript. Please see the revised manuscript.

8.In Figure.5a, the axes are wrong.

A8. The name of X-axis in Figure 5(a) has been corrected in revised manuscript.

9.In Figure.7, the expressions in the lower part of the graph are insufficient. for example, what is the a1? a2? a3?

A9. The supplementary notes have been provided in the revised version.

10.The interpretation of the data obtained in Figure.10 should be reconsidered. This is incomprehensible.

A10. The expression of interpretation of Figure 10 has been modified in revised manuscript. In this part, the distribution of instability region was mainly explained and the best processing range was proposed based on the higher value (mostly >0.5) of power dissipation energy and positive value of instability criteria. Similar interpretation can be found in the reference [19]. For the detailed of mechanism of dissipation energy efficiency, it was discussed in conjunction with Figure 12 in the last paragraph of Section 3.5

11.What is the difference between Figure 12b and 12c? What should be understood?

A11. The title names of the Figure 12(b) and 12(c) were written in the wrong order previously, and it has been corrected in the revised draft as follow:

Figure 12. Variations of power dissipation efficiency with (a) strain at 0.1 s-1, (b) temperature at strain rate of 0.1 s-1, and (c) strain at 1223K.

Figures 12(b) shows the variation of power dissipation efficiency with temperature under the different strain and the fixed strain rate of 0.1s-1. Figure 12(c) shows change of power dissipation efficiency under the different strain rate and the fixed temperature of 1223K. Since the power dissipation efficiency, η is related to microstructure evolution such as DRX, the discussion on the η value can help to deeply understand the deformation behavior and particularly judge the mechanism of dissipation energy either microstructural evolution dependent or temperature dependent. The value η ranges from zero to 1, and the higher value of η the better deformation capacity by microstructural evolution. For example, in this study, the value of η obtained in the deformation condition of 1123K/0.1s-1 shown in Figure 12(a) are mostly lower than 0.4 and it tends to decrease more with increase of the strain. This means the energy is mostly dissipated through heat by temperature rise effect, rather than by microstructural evolution. Considering these reasons, this study has discussed the variation of value of η under different deformation conditions. For clearer understanding, the relevant content has been rewritten. Please find the revised manuscript.

12.The conclusion should be expanded.

A12. Supplementary explanations for strain hardening exponent(n) and important finding of power dissipation energy mechanism change have been provided in the revised manuscript.

Reviewer 4 Report

This paper studies hot deformation behavior of the TiH2-based powder metallurgy Ti-1100 titanium alloy by the isothermal compression tests at temperatures of 1123-1323K, and strain rates of 0.01-1s-1. Strain hardening exponent (n) and strain rate sensitivity (m) were determined, in addition, the processing maps were constructed at the studied range of the temperature and strain rate for a strain of 0.6. The present paper is interesting, however, to be accepted for publication the following comments need to be addressed

-          Major English changes are required in the revised manuscript

Abstract

-          The abstract is well written.

Introduction

-          The introduction section needs to be improved.

-          L 37, the description of the Ti-1100 alloy should be modified from (Ti-/6Al-/2.75Sn-/4Zr-/0.4Mo-/0.4Si) to (Ti-6wt.%Al-2.75 wt.%Sn-4 wt.%Zr-0.4 wt.%Mo-0.4 wt.%Si).

Material and experimental details

-          The authors should add how many samples were used for isothermal compression test. At least three samples should be used for each condition for ensuring the repeatability and to know the uncertainty of the measured values.

-          Please rewrite the caption of Table 1, it is (Table 1. This is a table. Tables should be placed in the main text near to the first time they are cited.) is it correct?????

Results

-          It is recommended to add the side view for the samples in figure 1 to show the cracks clearly. L155, I can not see any cracks at this condition. Please show it by using drawing tools, modify the figure to support your claim.

-          L 197, what is the meaning of the “lip system”?

-          Another cause of the strain softening, and hardening for different ti alloys can be found in the paper, https://doi.org/10.3390/met8100819

Otherwise, the results are well presented and discussed.

Author Response

The authors appreciate very much for the Reviewer’s suggestions and comments which helped us to improve our manuscript. Those are very constructive and fruitful to enhance the quality of this manuscript. Please find our revised version of the manuscript. We have made some modifications according to the Reviewer’s suggestions and comments. Important modifications have been highlighted by blue color. A detailed description for each modification is presented in this letter.

This paper studies hot deformation behavior of the TiH2-based powder metallurgy Ti-1100 titanium alloy by the isothermal compression tests at temperatures of 1123-1323K, and strain rates of 0.01-1s-1. Strain hardening exponent (n) and strain rate sensitivity (m) were determined, in addition, the processing maps were constructed at the studied range of the temperature and strain rate for a strain of 0.6. The present paper is interesting, however, to be accepted for publication the following comments need to be addressed

-  Major English changes are required in the revised manuscript

A1.English expression and grammar have been checked and some revisions have been made through the whole manuscript.

Abstract

-  The abstract is well written.

Introduction

-  The introduction section needs to be improved.

A2. The introduction has been polished and some modification have been made as follow:

(1) Some expressions in the section paragraph of the introduction have been revised. Besides, the reference [12] which is not directly related to this study has been deleted. (2) Some expression in third paragraph of the introduction have been revised.

Please see the revised draft.

-   L 37, the description of the Ti-1100 alloy should be modified from (Ti-/6Al-/2.75Sn-/4Zr-/0.4Mo-/0.4Si) to (Ti-6wt.%Al-2.75 wt.%Sn-4 wt.%Zr-0.4 wt.%Mo-0.4 wt.%Si).

   A3. The typo has been corrected in revised manuscript.

Material and experimental details

-   The authors should add how many samples were used for isothermal compression test. At least three samples should be used for each condition for ensuring the repeatability and to know the uncertainty of the measured values.

A4. Due to limited time and resources, it is not possible to supplement the repeated experiments at this moment. Nevertheless, the authors believes that the experimental results are all reliable and accurate due to following reasons: (1) During the whole thermal compression process, the experiment was very stable. It can be known from the experimental raw data that the deformation rate (called stroke in the test) can reach a stable level of 10-3 order of magnitude (for example in the test of 1173K/0.01s-1), and no abnormal phenomenon or problems such as sample flying out between the dies were found during whole thermal compression test. (2)The thermal compression test was conducted in THERMECMASTER–Zthermos–simulation machine(Fuji, Japan), which is one of typical modern machines used in the field of thermal compression and no problem caused by equipment were found during test. Moreover, it is generally found in the literature, the thermal compression test is very stable and specific reproduced test provided.

-   Please rewrite the caption of Table 1, it is (Table 1. This is a table. Tables should be placed in the main text near to the first time they are cited.) is it correct?????

   A5. The title of Table 1 has been corrected in revised manuscript.

Results

-  It is recommended to add the side view for the samples in figure 1 to show the cracks clearly. L155, I can not see any cracks at this condition. Please show it by using drawing tools, modify the figure to support your claim.

    A6. Figure 1 has been updated and the side view for the samples was also included. Accordingly, the description of Figure 1 has been modified in revised manuscript.

-   L 197, what is the meaning of the “lip system”?

 A7. This was misspelling. The typo "lip" has been corrected to "slip"

-   Another cause of the strain softening, and hardening for different ti alloys can be found in the paper, https://doi.org/10.3390/met8100819

Otherwise, the results are well presented and discussed.

A8. The mentioned paper presents superplastic deformation behavior by using the tensile test of different α+β type of titanium alloys. In mentioned paper, the mechanism of strain hardening/softening effect considered in tensile test is basically different from that considered in the hot compression test of this paper. In mentioned paper, the strain softening is the results of dynamic recrystallization and the strain hardening is explained by the dynamic growth. Differently, in the compression test of our work and relevant works [28,30,31], the softening effect is generally caused by ether dynamic recrystallization or temperature rise effect as discussed in section 3.2. For strain hardening effect, it is generally believed that strain hardening is mainly caused by dislocations proliferate and intersect with increase of strain during compression [23]. Therefore, the cause of strain softening/hardening mentioned in the above paper is not considered in this study.

Round 2

Reviewer 1 Report

I recommend paper for publication in present form.

Reviewer 3 Report

This paper is sufficient for the acceptance.

Reviewer 4 Report

The authors have addressed all the comments, the revised manuscript can be accepted for publication in the current form.